# Relationship between Southern Hemispheric jet variability and forced response: the role of the stratosphere

Philipp Breul[1], Paulo Ceppi[1,2], and Theodore G. Shepherd[3]

[1]Department of Physics, Imperial College London, London, United Kingdom
[2]Grantham Institute, Imperial College London, London, United Kingdom
[3]Department of Meteorology, University of Reading, Reading, United Kingdom

**Correspondence:** Philipp Breul (pyb18@ic.ac.uk)

**Abstract.** Climate models show a wide range of Southern Hemispheric jet responses to greenhouse gas forcing. One approach to constrain the future jet response is by utilising the fluctuation-dissipation theorem (FDT) which links the forced response to internal variability timescales, with the Southern Annular Mode (SAM) the most dominant mode of variability of the Southern Hemispheric jet. We show that interannual stratospheric variability approximately doubles the SAM timescale during austral summer in both re-analysis data and models from the Coupled Model Intercomparison Project, phases 5 (CMIP5) and 6 (CMIP6). Using a simple barotropic model, we demonstrate how the enhanced SAM timescale subsequently leads to an overestimate of the forced jet response based on the FDT, and introduce a method to correct for the stratospheric influence. This result helps to resolve a previously identified discrepancy between the seasonality of jet response and the internal variability timescale. However, even after accounting for this influence, the SAM timescale cannot explain inter-model differences in the forced jet shift across CMIP models during austral summer.

## 1 Introduction

Global climate models (GCMs) generally predict a poleward shift of the Southern Hemispheric eddy-driven jet in response to greenhouse gas forcing, but the magnitude of this shift is highly uncertain (Barnes and Polvani, 2013; Curtis et al., 2020). This in turn has consequences for the predictability of climate change impacts on the mid-latitude regions (Shepherd, 2014). It is therefore desirable to constrain the range of future jet responses.

One way of constraining forced responses in a system is by utilising the fluctuation-dissipation theorem (FDT), first introduced into the climate community by Leith (1975), which links internal variability timescales to the forced response (Gritsun and Branstator, 2007; Ring and Plumb, 2008). For the Southern Hemispheric jet, the Southern Annular Mode (SAM) is the leading mode of internal variability, which constitutes a north-south shift of the jet. A very simplified version of the FDT linearly relates the SAM timescale to the forced response of that mode (see Section 2) and has been used in previous studies to interpret externally-forced responses of the SAM and the jet stream (e.g. Gerber et al., 2008b; Kidston and Gerber, 2010).

In an inter-model comparison, Kidston and Gerber (2010) found correlations between the timescale of the SAM, the climatological jet latitude, and the forced jet shift in the Southern Hemisphere, which they interpreted using FDT arguments. However, Simpson and Polvani (2016) cast doubt on these findings after considering the seasonality of these relationships:

inter-model differences in SAM timescale occur mainly in austral summer, whereas the relationship between climatological jet latitude and jet shift is found primarily in austral winter – when spread in SAM timescale is minimal. Thus, the applicability of the FDT to the Southern Hemispheric jet response to forcing remains unclear.

     One explanation for the findings of Simpson and Polvani (2016) could be that the inter-model differences in SAM timescale in austral summer do not reflect differences in internal variability persistence. There is a distinct increase in SAM timescale

during summer, both in observations and in models, and several studies have suggested a stratospheric influence on the jet to be at least partially responsible for this phenomenon (Gerber et al., 2010; Simpson et al., 2011). A possible mechanism was proposed by Byrne et al. (2017), who demonstrated that the stratospheric polar vortex breakdown (VB) induces an equatorward regime transition of the tropospheric eddy-driven jet. They proposed that interannual variations in VB timing may consequently enhance the SAM persistence during austral summer by inducing non-stationarity in the zonal wind signal. This would con-

found any estimate of the timescale of internal SAM variability, when treating the stratospheric influence as exogenous to the troposphere – an assumption which will be discussed further below.

     Here we demonstrate the impact of stratospheric vortex variability on the tropospheric jet, both in terms of unforced interannual variability, and the forced response in 21$^{st}$-century scenarios. In the first part of this paper, we demonstrate how externally-induced variability (like the interannual variation in VB date) inflates the SAM timescale, and introduce a method

to correct for this effect. The second part of the paper then considers the implications of these results for the prediction of the forced jet response when using the FDT.

## 2    Fluctuation-Dissipation Theory

The fluctuation-dissipation theorem links internal variability to the forced response of a system, enabling the prediction of a response to forcing by observing the unperturbed system. This makes an application to the problem of climate change very

attractive. Several versions of the FDT for the climate system have been proposed in the past with differing assumptions (e.g., Leith, 1975; Gritsun and Branstator, 2007; Ring and Plumb, 2008). The FDT has been applied to climate models of various degrees of complexity, but with mixed results. Some studies reported relatively good skill (Gritsun and Branstator, 2007; Fuchs et al., 2015; Gritsun and Branstator, 2016), while others found the predictions to match only qualitatively (Ring and Plumb, 2008) or only for a subset of forcings (Lutsko et al., 2015; Hassanzadeh and Kuang, 2016). A brief discussion on previous

literature on FDT shortcomings in the climate system can be found in Appendix A.

     Here we use an FDT formulation based on Gritsun and Branstator (2007), which we translat into EOF space (denoted by $\hat{\cdot}$); for more information see Appendix B. The FDT relation becomes

$$\delta\hat{\boldsymbol{u}} = \hat{L}\,\delta\hat{\boldsymbol{f}}, \tag{1}$$

with $\delta\hat{\boldsymbol{u}} = \mathbb{E}[\hat{\boldsymbol{u}}' - \hat{\boldsymbol{u}}]$ the response as the difference of the perturbed and unperturbed state vectors, $\delta\hat{\boldsymbol{f}}$ the forcing and $\hat{L}$ the

FDT matrix, which can be calculated from internal variability of the unforced system. For more information on the response matrix $\hat{L}$, we refer again to Appendix B where we also discuss the advantages of considering the relation in EOF space.

From Eq. (1) we can isolate the response projection onto EOF1 by considering only the first vector entry

$$\delta \hat{u}_1 = \sum_{i=1}^{n} \hat{L}_{1,i} \delta \hat{f}_i, \qquad (2)$$

where $\delta \hat{u}_1$ is the EOF1 response and $\delta \hat{f}_i$ the forcing of the $i$-th EOF. We note that the entry $\hat{L}_{1,1}$ is the integral timescale of the first principal component. A commonly used approximation of Eq. (2) is obtained by dropping all terms that do not contain $\hat{L}_{1,1}$ to get the one-dimensional relation

$$\delta \hat{u}_1 \approx \lambda_{1,1} \delta \hat{f}_1, \qquad (3)$$

where $\lambda_{1,1}$ is equal to the matrix entry $\hat{L}_{1,1}$; for details, see Appendix B. However, the validity of this approximation depends on $\lambda_{1,1} \delta \hat{f}_1$ being much larger than the other terms in Eq. (2). This means we require the forcing to project strongly onto EOF1 and/or $\lambda_{1,1}$ to be large relative to the other entries in the response matrix $\hat{L}$. The approximation in Eq. (3) can therefore, depending on the situation, be an oversimplification; a related point was demonstrated by Hassanzadeh and Kuang (2016).

As an aside, we note that the timescale $\lambda_{1,1}$ in Eq. (3) is formally defined as the integral timescale, as detailed in Appendix B. When the lagged decorrelation decreases exponentially, however, the integral timescale is identical to the $e$-folding timescale. Except when making FDT predictions in the barotropic model (see Section 4.3) we therefore use the $e$-folding timescale as a measure for SAM persistence, since it is often used in the literature and we found it to be less noisy than the integral timescale when using a limited amount of data. Note however that the results presented here do not differ qualitatively when using the integral timescale instead.

Results similar to Eq. (3) are the physical grounding for trying to identify emergent constraints. When applied to the eddy-driven jet, $\lambda_{1,1}$ becomes the SAM timescale and has thus been proposed as a potential constraint on the forced response of the SAM or, equivalently, the midlatitude jet shift (e.g., Kidston and Gerber, 2010; Simpson and Polvani, 2016; Gerber et al., 2008a). This would be useful especially for the CMIP models where not enough data is available to resolve the full correlation structure.

The above-mentioned studies did not take into account stratospheric effects, but based their analysis solely on tropospheric variables (as is common in the literature). It has however been shown that the stratospheric polar vortex exerts a strong influence on the eddy-driven jet when it breaks down (Byrne et al., 2017), inducing regime transitions. While most of the stratospheric variability ultimately comes from the troposphere, the stratospheric processes happen on much longer timescales. Even a stratospheric ENSO pathway has been identified (Byrne et al., 2019; Kretschmer et al., 2021), with ENSO acting on multi-year timescales. Therefore we make the assumption that the stratosphere acts as an exogenous forcing on the troposphere on timescales of SAM variability. This is similar to the common assumption that sea-surface temperature anomalies can force the atmospheric winds, even though the sea surface temperatures themselves also integrate the atmospheric forcing (Barsugli and Battisti, 1998). We thus hypothesise that the increased SAM timescale during austral summer is not reflecting increased persistence owing to internal tropospheric dynamics, but instead that it is due to the stratospheric influence. We aim to investigate to what extent internal variability is influenced by the stratosphere and whether an emergent constraint between SAM timescale and forced response can be found once this influence is accounted for.

# 3 Data and Methods

## 3.1 Barotropic Model

We conduct idealised experiments with a simple stochastically-forced barotropic model, first introduced by Vallis et al. (2004). We direct the reader there for details, but discuss the key features of the model and experimental setup here. The model integrates the single-layer barotropic vorticity equation on the sphere with added linear drag $r$, hyperdiffusion $\nabla^4 \zeta$ and a
 random wavemaker $S$:

$$\frac{D\zeta}{Dt} = S - r\zeta - \kappa\nabla^4\zeta. \tag{4}$$

The wavemaker excites Rossby waves in the mid-latitudes and this setup is sufficient to lead to the formation of an eddy-driven jet despite the absence of baroclinic instabilities. In all our experiments, we use a T42 grid and a timestep of 1800 s. We set the linear drag to $r = (6.5\,\mathrm{days})^{-1}$, and $\kappa$ is resolution dependent to remove enstrophy at small scales. The stirring $S$ is a
 random process that excites total wavenumbers 8–12, restricted to zonal wavenumbers greater than 3. The stirring strength in each wavenumber varies randomly between $(-A, A) \times 10^{-11}$, with $A = 9.0\,\mathrm{s}^{-2}$. Additionally, to confine $S$ meridionally we multiply it by a latitudinal Gaussian window centred at $-40°$ with a standard deviation of $12°$. The stirring has a temporal decorrelation timescale of 2 days. Note that the setup used here is the same as the BARO setup of Barnes and Thompson (2014). All experiments are spun up for 500 days and then run for another 150 days, with 2000 ensemble realisations per experiment.
 To parameterise the influence of the polar vortex on the jet, we use a torque that mimics the first (unweighted) EOF of the BARO setup and thus leads to a poleward shifting of the jet. We use the unweighted EOF since it makes the FDT representation mathematically easier. While it is also possible to use weighted EOFs in the FDT, as discussed in Appendix B, for the data used here the weighted and unweighted EOF1 only differ marginally, as can be seen in Fig. 1. The shape of the torque, shown as the dotted line in Fig. 1, is chosen to approximate EOF1:

 $$f_{\mathrm{eof}} = a \cdot \sin(3.5\theta') \cdot e^{-\left(\frac{\theta'}{0.3}\right)^2 + \frac{\theta'}{0.7}}, \quad \theta' = -2\pi(\theta + 40.2°)/360°, \tag{5}$$

with $\theta$ the latitude in degrees and $a$ the amplitude of acceleration. We set the amplitude to $a = -2\,\mathrm{m\,s}^{-1}\,\mathrm{day}^{-1}$, leading to a poleward shift when the torque is active. While this torque is not designed to accurately represent the upper-tropospheric zonal wind forcing associated with the stratospheric vortex, the chosen torque amplitude produces a jet shift similar in magnitude to that observed in ERA5 following the VB (Fig. 2). Note, however, that the exact value of $a$ is not important qualitatively for the
 results shown here. In fact the exact structure of the forcing should be of limited importance as long as it produces a shift of the jet.

We perform four experiments (Table 1): OFF, a control experiment with no torque; ON, with the torque active throughout the run; and two more experiments starting with an active torque, which is then switched off at some point during the run to mimic the vortex breakdown. For the first of these experiments, VB-FIX, the switch-off always happens on day 50 (not counting the
 spin-up period) to create a setup without interannual variability in VB date. In the second experiment, VB-VAR, we mimic the ERA5 interannual variability by switching off the torque at a varying time between day 25 and 75, uniformly distributed

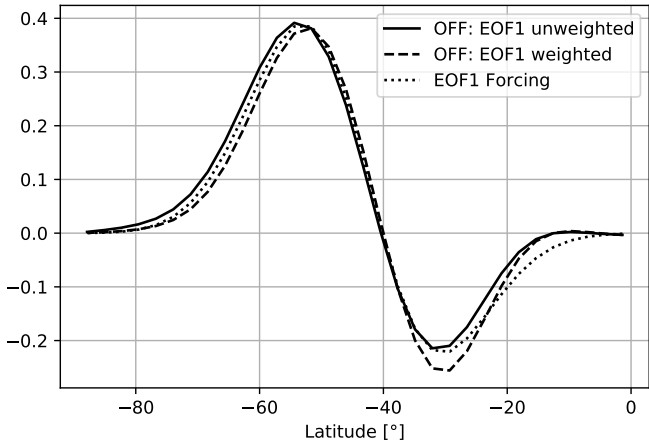

**Figure 1.** The EOF of zonally averaged zonal wind of the barotropic OFF experiment, both with and without using a $\sqrt{\cos(\text{lat})}$ weighting. Additionally shown is the torque forcing of Eq. (5) which aims at approximating the first unweighted EOF. All curves are normalized by their Euclidean norm.

among realisations. (Although a uniform distribution may not be realistic, we do not expect the results to be overly sensitive to the choice of distribution.) In both VB-FIX and VB-VAR the switch-off does not happen instantaneously in time: we smear out the transition using $0.5[1 - \tanh((t_0 - t)/3))]$, with the time $t$ in days, and $t_0$ the vortex breakdown day. This means it takes about 14 days for the torque to transition from more than 99% to less than 1% of its strength.

Additionally, we perform forced experiments in the VB-FIX and VB-VAR setups (Table 1), using two types of forcings. One is a Gaussian torque centred at $55°$S with a standard deviation of $5°$ and a maximum amplitude of $0.5 \text{ m s}^{-1} \text{ day}^{-1}$; these experiments are referred to as VB-FIX-Gauss-FOR and VB-VAR-Gauss-FOR. The other forcing uses the EOF1-like torque function from Eq. (5) with an amplitude of $a = 0.75 \text{ m s}^{-1} \text{ day}^{-1}$, and the experiments are called VB-FIX-EOF1-FOR and VB-VAR-EOF1-FOR. The numerical response is calculated as the difference in zonally-averaged zonal wind between forced and unforced experiments, averaged over all experiment realisations.

### 3.2 ERA5, CMIP5 & CMIP6

To compare to real-world data, we use European Centre for Medium Range Weather Forecasts Reanalysis 5 (ERA5; Hersbach et al., 2020) 1950–2020 daily instantaneous data, sampled at 12pm UTC. We exclude year 2002, as did Byrne et al. (2017), since this year includes a sudden stratospheric warming event that has a large influence on our VB day index, but note that our results do not change qualitatively when including this year.

To compare our idealised barotropic experiments to more complex models, we use the historical and RCP8.5 experiments of the Coupled Model Intercomparison Project phase 5 (CMIP5), and the historical and SSP5-8.5 experiments of the Coupled Model Intercomparison Project phase 6 (CMIP6). We define the response as the difference between years 2080–2099 of RCP8.5/SSP5-8.5 and the 1950–2005/1950–2013 climatology from the historical runs. We use the historical runs rather

| Unforced Experiments | Polar Vortex Breakdown Date | Forcing |
|---|---|---|
| OFF | always off | - |
| ON | always on | - |
| VB-FIX | day 50 | - |
| VB-VAR | varies: day 25–75 | - |
| Forced Experiments | | |
| VB-FIX-EOF1-FOR | day 50 | EOF1 |
| VB-FIX-Gauss-FOR | day 50 | Gaussian |
| VB-VAR-EOF1-FOR | varies: day 25–75 | EOF1 |
| VB-VAR-Gauss-FOR | varies: day 25–75 | Gaussian |

**Table 1.** Barotropic experiment setups. All unforced experiments share the same basic setup described in the text (identical to BARO in Barnes and Thompson, 2014) and only differ in the influence of the stratospheric polar vortex, which is parameterised here as a torque given by Eq. (5). OFF and ON are control experiments, mimicking an absent or active polar vortex, respectively. VB-FIX and VB-VAR mimic a vortex breakdown respectively without and with interannual variability. The latter two setups are used to perform forced experiments, applying a Gaussian forcing centred at $55°$S with a standard deviation of $5°$ and a maximum amplitude of $0.5 \, \mathrm{m \, s^{-1} \, day^{-1}}$ (suffix Gauss-FOR), or an EOF1-like forcing from Eq. (5) with $a = 0.75 \, \mathrm{m \, s^{-1} \, day^{-1}}$ (suffix EOF1-FOR).

than piControl owing to the larger availability of daily instantaneous data, but note that our results are qualitatively similar when using piControl instead, which shows that ozone depletion in the historical period does not substantially influence the results. We include the following 22 CMIP5 models: ACCESS1.0, ACCESS1.3, BCC-CSM1.1, BNU-ESM, CanESM2, CMCC-CESM, CMCC-CM, CMCC-CMS, CSIRO-Mk3.6.0, EC-EARTH, FGOALS-g2, GFDL-ESM2G, GFDL-ESM2M,
IPSL-CM5A-MR, IPSL-CM5B-LR, MIROC5, MIROC-ESM-CHEM, MIROC-ESM, MPI-ESM-LR, MPI-ESM-MR, MRI-CGCM3, and NorESM1-M; and the following 20 CMIP6 models: ACCESS-CM2, CanESM5, CESM2-WACCM, CMCC-CM2-SR5, CMCC-ESM2, CNRM-CM6-1, EC-Earth3-CC, FGOALS-g3, HadGEM3-GC31-LL, HadGEM3-GC31-MM, IITM-ESM, INM-CM5-0, KACE-1-0-G, MIROC6, MPI-ESM1-2-HR, MPI-ESM1-2-LR, NorESM2-LM, NorESM2-MM, TaiESM1 and UKESM1-0-LL.

All results are based on the Southern Hemispheric zonally-averaged zonal winds between $0°$ and $78°$S. We use the wind at 850 hPa as a representation of the tropospheric jet and wind at 50 hPa and $60°$S latitude as indicating the stratospheric polar vortex strength. As in Ceppi and Shepherd (2019), we define the vortex breakdown day as the final time when the stratospheric polar vortex strength drops below 15 m/s in late austral spring to early summer. Furthermore, we define the jet latitude as the maximum of the parabola fitted to the maximal zonal wind value and its two adjacent entries. Global warming was calculated
as the difference in the cos(lat) weighted spatial and temporal average of the temperature at the surface between the periods 1950–2005 and 2080–2099 for CMIP5, and between 1950–2013 and 2080–2099 for CMIP6.

To calculate the SAM timescale, we first deseasonalise the zonally averaged zonal wind and then perform an EOF decomposition with the standard $\sqrt{\cos(\text{lat})}$ area weighting. We then calculate the $e$-folding timescale of the lagged autocorrelation function of the first principal component for every day of the year to get the time-resolved SAM timescale. Note that we will later introduce a slightly altered method to take the stratospheric interannual variability into account.

## 4    Results & Discussion

### 4.1    Internal Variability in ERA5 & the Barotropic Model

We first consider the ERA5 climatological jet position (black curve in Fig. 2a), which shows that the jet transitions between a more equatorward position in the winter and summer seasons and a more poleward position in the spring and autumn. We are particularly interested in the equatorward transition in late austral spring to early summer, which coincides with the interannual range of VB dates (denoted by the dashed part of the black curve in Fig. 2a). We observe that in years with earlier than average VB, the jet transitions equatorward earlier than in years with late VB (Fig. 2a, blue and green curves). This is consistent with the findings of Byrne et al. (2017) that the equatorward jet transition in early austral summer is a direct consequence of the polar stratospheric VB.

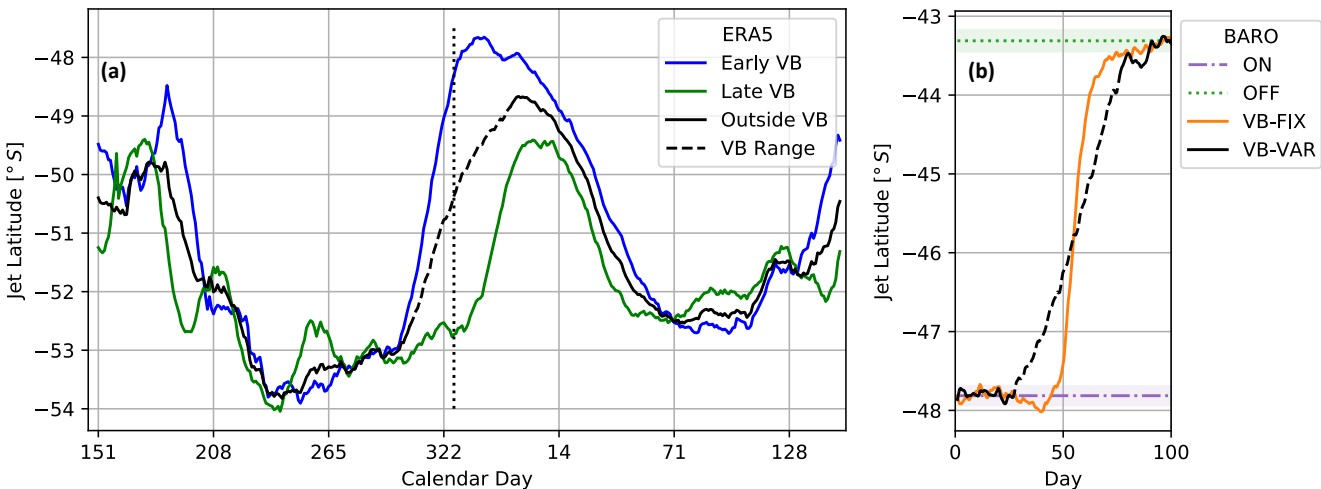

**Figure 2.** Climatological jet position. **(a)** Jet latitude in ERA5 based on zonal-mean zonal wind at 850hPa, smoothed with a 20-day running window. Shown in black is the climatology with the dashed part constituting the range of interannual vortex breakdown days which coincides with the equatorward jet shift in late austral spring to summer. The early and late VB years show an early and late jet shift respectively. The dotted vertical line represents the mean VB day. **(b)** Jet latitude in a simple barotropic model mimicking the late spring to austral summer transition, which is sharper in the absence of interannual VB variability (VB-FIX) compared with the varying case (VB-VAR).

Coincident with the VB period, we also find a substantial increase in SAM timescale (Fig. 3a, dashed black curve), peaking at around 20 days in late November; by comparison, the timescale remains below 10 days outside of the November to January

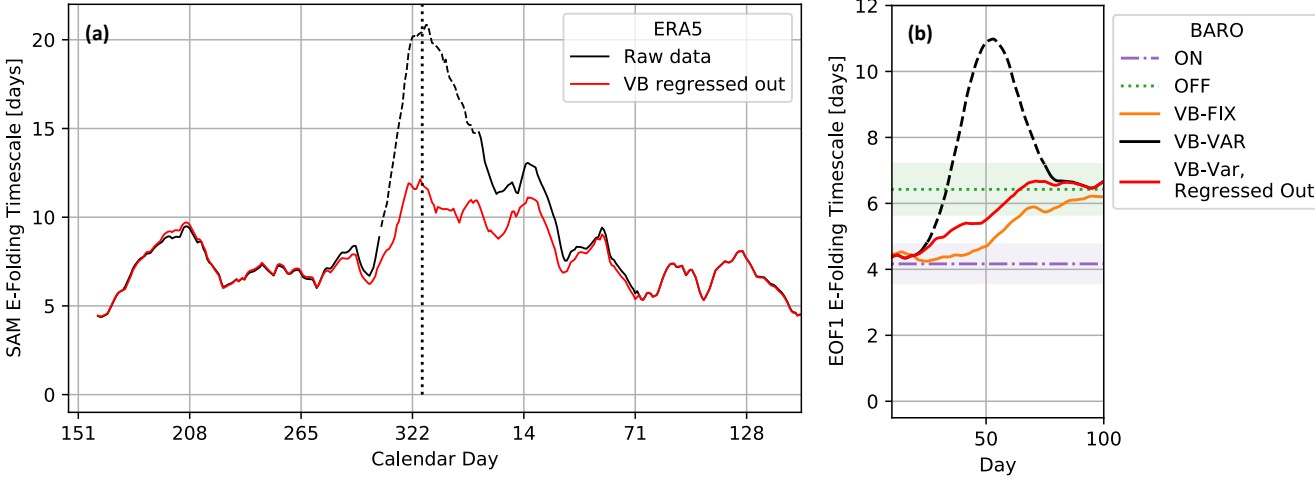

**Figure 3.** *e*-folding timescales of the first EOF for the setups shown in Fig. 2. The timescales have been averaged with a 20-day running window. **(a)** ERA5; the black curve shows the *e*-folding timescale calculated from raw data, with the dashed part denoting the range of VB days. The red curve shows the *e*-folding timescale calculated from data where the VB influence was regressed out. **(b)** Barotropic model results; same as in (a) for VB-VAR, additionally the orange curve shows the *e*-folding timescale of the VB-FIX experiment. The horizontal dotted and dash-dotted lines indicate the EOF1 timescales in the unperturbed ON and OFF experiments.

period. This peak in SAM timescale during late spring to early summer has been noted in several prior studies (e.g., Gerber et al., 2008a, 2010; Simpson et al., 2011). We ascribe this timescale enhancement to interannual variability in the stratospheric VB timing: when deseasonalising the data with the daily climatology over all years, we introduce non-stationarity since the jet

transitions earlier or later than average, depending on the VB date (Byrne et al., 2017).

Now turning to the barotropic experiments, we observe a quantitatively similar jet shift in the VB-FIX and VB-VAR experiments relative to ERA5 (Fig. 2b). We also note that the mean jet position is biased about $5°$ equatorward compared to ERA5, which is unsurprising since we do not tune the stirring latitude to approximate the ERA5 climatology as closely as possible. As expected, the VB-VAR climatology shows a substantially shallower equatorward jet transition compared with VB-FIX.

The impact of variability in the VB timing in the barotropic experiments (as simulated by the variability in torque forcing in VB-VAR) is demonstrated in Fig. 3b, where the SAM timescale peaks at approximately twice its unperturbed value around day 50 of the experiment (dashed black curve). While the absolute increase in SAM timescale is smaller in the barotropic model compared to ERA5, the relative increase is similar. The smaller SAM timescales in the barotropic model are likely due to the lack of baroclinic feedback in this model (Barnes and Thompson, 2014). Meanwhile, the VB-FIX experiment only shows a

smooth transition between the two ON and OFF equilibrium states (Fig. 3b, orange curve). From past studies we know that SAM persistence tends to increase as the mean jet latitude decreases (Kidston and Gerber, 2010; Barnes and Hartmann, 2010), which likely explains the change in SAM timescale between the two equilibrium states. Since the only difference between the

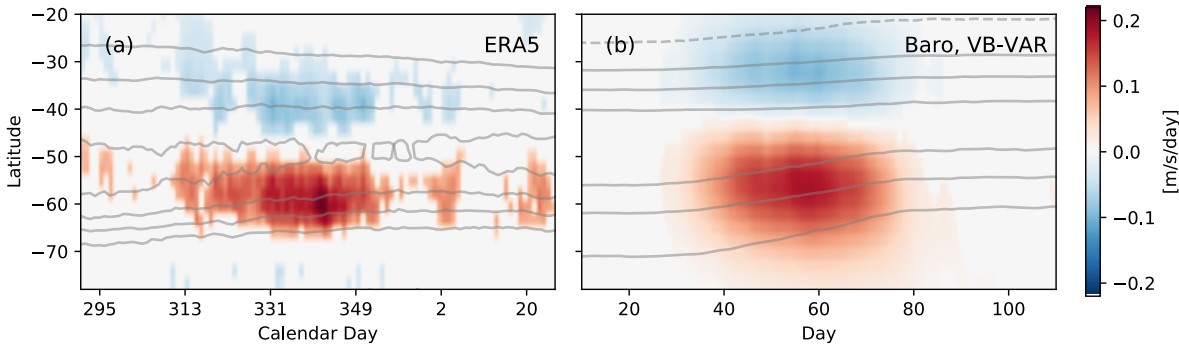

**Figure 4.** Zonally-averaged zonal wind anomalies regressed onto VB date anomalies for each latitude and day independently for **(a)** ERA5 and **(b)** the barotropic VB-VAR experiment. Entries that were not statistically significant at the 99% level (based on a Student's t-test) were set to zero for visualisation.

VB-FIX and VB-VAR experimental setups lies in the variability in VB dates, the results in Fig. 3b unequivocally demonstrate how such variability can inflate SAM timescales.

As an aside, we note that Fig. 2 seems to suggest that the amount of jet shift may depend on the VB date, with early VB years seeing a larger equatorward shift. In the barotropic setup shown here, we have not included this effect for reasons of simplicity. Including this effect would only inflate the SAM timescale further, by making the zonal wind even less stationary.

The results so far suggest that the enhanced SAM persistence in early austral summer is attributable (at least in part) to VB variability. If this is the case, then it should be possible to correct for this effect, provided the impact of the VB variability on

the jet latitude is known. To estimate this impact, for both ERA5 and the barotropic model data we regress the zonally averaged zonal wind anomalies at every point in latitude and time separately onto the VB date anomalies (Fig. 4). In both cases the regressions show a dipole pattern centred around the jet latitude during the VB period, confirming that variability in VB date is associated with a jet shift.

We then use these regression maps to regress out the stratospheric influence for each year (or ensemble member in the

barotropic experiments) according to the respective VB dates. While this method assumes a linear relationship between VB date and jet latitude anomaly, we tested both non-linear regression methods (quadratic and piecewise-linear) and binning by VB day as an alternative way to account for the stratospheric influence. This yielded qualitatively similar results for both the barotropic model and ERA5. We also note that our measure of vortex variability is inevitably somewhat arbitrary, and other indices could be regressed on instead – for example the anomalous zonally averaged zonal wind at $50$ hPa and $60°$S latitude,

in which case a lag time has to be chosen. When making this analysis we do not find the overall results to change qualitatively.

Applying this regression technique to the VB-VAR barotropic experiment, we recover an estimate of the internal variability timescale free from stratospheric influence, shown in red in Fig. 3b. This "corrected" SAM timescale does not show a peak, but instead transitions mostly monotonically between the ON and OFF states, similar to the VB-FIX experiment. (Note that we do not expect the red and orange curves in Fig. 3b to be strictly identical, since the climatological transition between the

two equilibrium states is more gradual in the VB-VAR case.) Doing the same with ERA5 data (red curve in Fig. 3a), we see that most of the summertime increase in SAM timescale vanishes. We speculate that the remaining increase may be associated with jet latitude changes (Kidston and Gerber, 2010; Barnes and Hartmann, 2010), since the jet is positioned more equatorward during the time of increased corrected SAM persistence.

## 4.2    Internal Variability in CMIP5&6

To determine the implications of our results for climate models, we turn to the historical CMIP5 and CMIP6 GCM experiments, and apply the same regression technique described above to quantify and correct for the VB influence on the SAM timescale (Fig. 5a and c respectively). Both the CMIP5 and CMIP6 model means show a clear decrease in SAM timescale, similar to what was observed in modelling experiments that aimed at taking out the stratospheric influence (Simpson et al., 2011). Considering the individual models, each represented by a thin line, we see that some have very pronounced peaks in SAM timescale during
late austral spring to summer. Differences in the timing of these peaks are likely attributable to inter-model differences in the VB timing (Wilcox and Charlton-Perez, 2013; Ceppi and Shepherd, 2019). When regressing out the VB influence, these peaks vanish or are substantially reduced. This shows that the very long SAM timescales simulated by some models are at least partly a result of stratospheric influence, rather than reflecting true internal tropospheric variability.

We note that unlike ERA5, the CMIP5 model mean still shows a pronounced peak of corrected SAM timescale in austral
summer; individual models even show peaks of up to about 40 days even after accounting for the VB effect (Fig. 5a). While we found a slight dependence of the magnitude of this effect on the model climatology, with equatorward biased models tending to show a stronger peak (not shown), further work would be needed to determine the exact cause. While still present, the residual peak is significantly reduced in the CMIP6 models (Fig. 5c). However, the VB effect appears much more similar between CMIP models and ERA5 when considered relative to the uncorrected SAM timescale (Fig. 5b and d), with the timescale being
approximately halved by accounting for VB variability in all cases. This result supports the reasoning that the leftover peak in SAM timescale around calendar day 1 in Fig. 5a (and to a lesser extent in Fig. 5c) is unrelated to the vortex breakdown. The conclusion is further supported by a modelling study from Simpson et al. (2011), which found a similar residual peak in SAM timescale after suppressing interannual variability in the stratospheric zonal-mean flow (green shading in their Fig. 3c), suggesting this feature is indeed physical.

As an aside, when comparing the results of CMIP5 and CMIP6 in Fig. 5 we find that the ERA5 SAM timescale is closer to the CMIP6 model mean than the CMIP5 model mean, both for the raw and corrected SAM timescale. Furthermore the spread in individual peaks in SAM timescale is clustered more closely around the ERA5 peak, resulting in a sharper model mean peak. This suggests that the CMIP6 models have an improved representation of stratosphere-troposphere interaction and possibly also tropospheric dynamics, but further analysis is needed to determine the exact cause of the improvements.

## 4.3    Forced Response in the Barotropic Model

Next we consider how the stratospheric impact on SAM timescales affects FDT predictions of the jet response to external forcing. Using the barotropic VB-FIX and VB-VAR setups, we perform two forced experiments (Section 3.1 and Table 1).

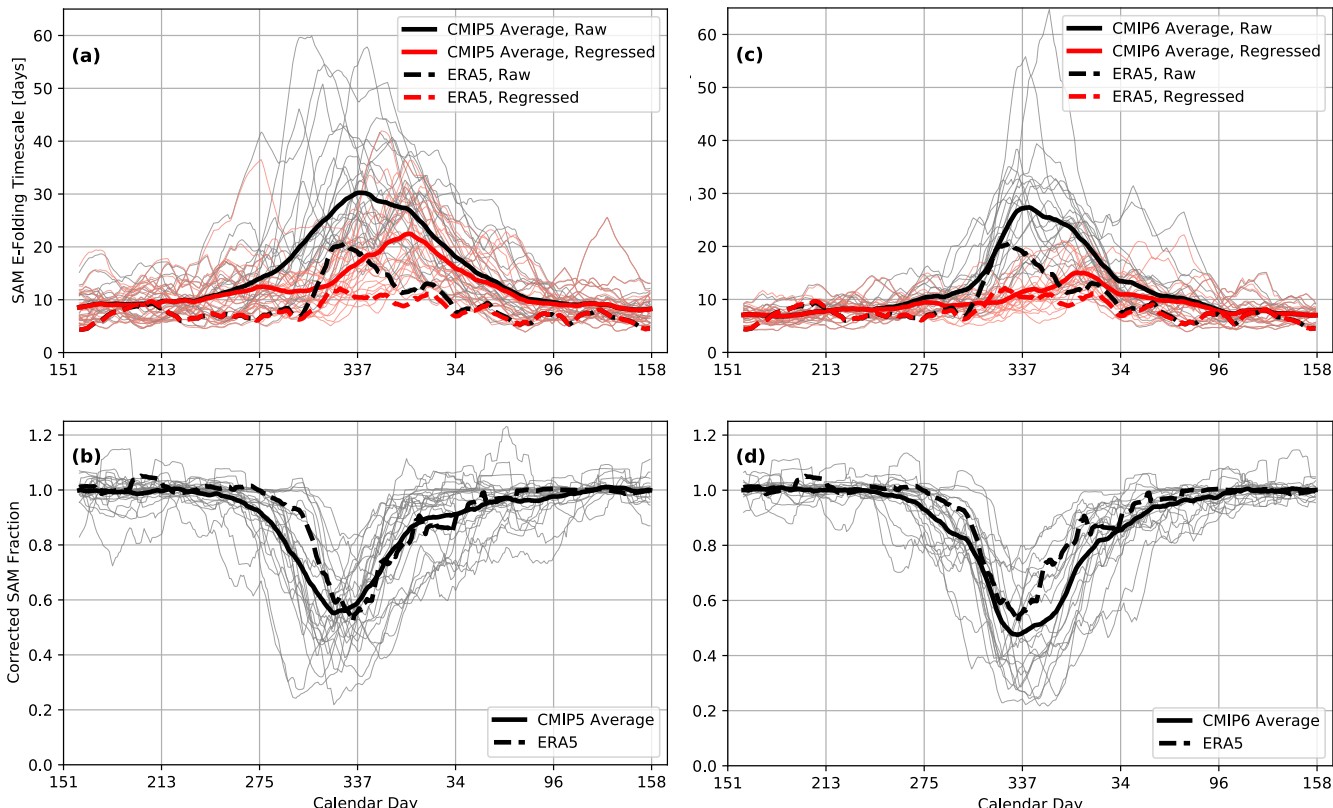

**Figure 5. (a)** Annual cycle of the corrected and uncorrected SAM e-folding timescale in CMIP5 simulations (black and red lines respectively). All results were calculated using zonally-averaged zonal wind at 850 hPa from the 1950–2005 historical experiments. The timescale was then averaged with a 20-day running window. Thin lines represent individual models. **(b)** Fraction of corrected to uncorrected SAM timescale. Grey lines show the individual CMIP5 models, with the solid black line the average over all grey curves; the dashed black line shows ERA5. **(c–d)** Same as (a–b) but for the 1950–2013 historical CMIP6 experiments.

In Figs. 6a–b we compare the numerical responses (we only show the numerical response for the VB-VAR and not for VB-FIX, since the two results were almost identical) to the predictions made using the full FDT method described in Eq. (1), which requires calculating the response matrices $\hat{L}$ from the VB-FIX and VB-VAR experiments (Fig. B1a–b). We calculate these predicted responses using the first 10 EOFs of zonally averaged zonal wind, which is sufficient since the forcing almost exclusively projects onto those (see Appendix B for further details).

For VB-FIX we find the FDT prediction to match the true response extremely well for both forcing cases – for VB-FIX-EOF1-FOR so closely that the lines are essentially indistinguishable (Fig. 6a, dashed orange and solid black curves). By contrast, for VB-VAR we observe a large overprediction (black curves): we interpret this as being due to the overestimation of the internal variability timescale, especially that associated with EOF1, as a consequence of the non-stationarity. This overprediction can however be corrected for by regressing out the VB influence (red curves in Fig. 6). This shows that regressing

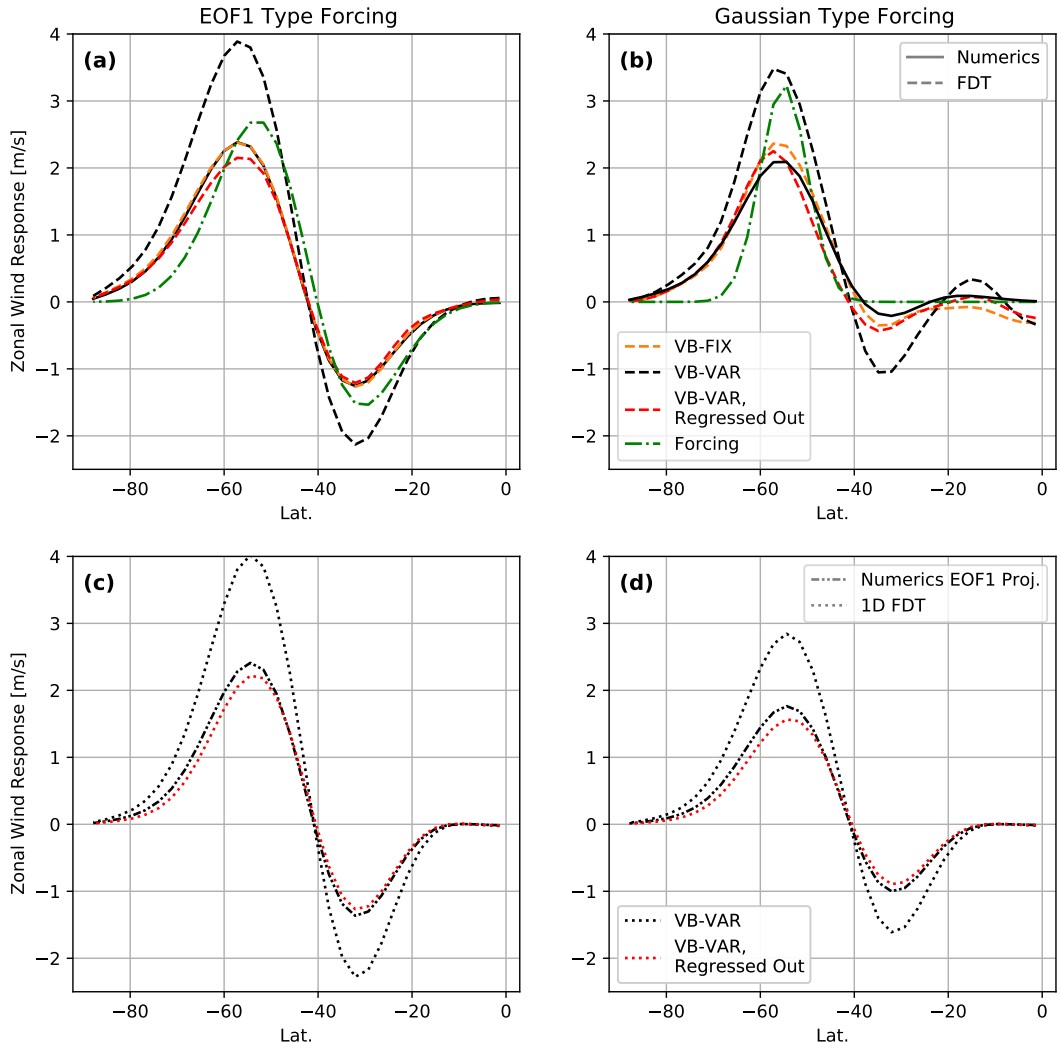

**Figure 6.** Response to two different types of forcing in the barotropic experiments VB-FIX and VB-VAR: **(a)** an EOF1-type forcing (experiments VB-FIX-EOF1-FOR and VB-VAR-EOF1-FOR), and **(b)** a Gaussian-type forcing (experiments VB-FIX-Gauss-FOR and VB-VAR-Gauss-FOR). The numerical responses are shown as solid lines and the FDT predictions (using the full method described in Appendix B) as dashed lines. Since the numerical responses of VB-FIX and VB-VAR are almost identical, we omitted the former. **(c–d)** Same as (a–b), but we compare the EOF1 projection of the numerical response with the 1D FDT predictions made using Eq. (3).

out the VB variability leads to better estimates of internal variability timescales (particularly the SAM timescale) and thus to more accurate FDT predictions in the barotropic model.

To obtain the results shown in Figs. 6a–b we used the full FDT method of Eq. (1). Most of the time this will not be an option for complex GCMs due to limited data availability and we have to revert to using the simple FDT relation of Eq. (3). To gauge how well the simple FDT relation holds for the experiments presented here, we compare the EOF1 projection of the numerical responses to the predictions made using Eq. (3) – see Figs. 6c–d. Again, we find that the VB-VAR experiment over-predicts the response in both forcing cases and that a better prediction (although slightly too weak) is recovered when regressing the VB

influence out. While the success of this method will most likely not translate directly to reanalysis data or GCMs, the results presented so far give confidence in the approach.

## 4.4   Forced Response in CMIP5&6

Next we assess the implications of our results for climate change projections. It is presently unclear whether the SAM timescale can provide a useful constraint on the jet response to external forcing in complex GCMs. Revisiting the relationship identified

by Kidston and Gerber (2010), Simpson and Polvani (2016) found no relationship between SAM timescale and forced jet shift across CMIP5 models during December–January–February (DJF; their Fig. 2f). Our results so far indicate that the raw SAM timescale is not suited for FDT predictions, being inflated by the effect of VB variability. We therefore consider whether the SAM timescale may constrain the DJF jet response better once the VB effect has been accounted for – assuming that the stratospheric effect is exogenous to the troposphere as discussed in Section 2.

As in prior studies, we apply the approximated one-dimensional FDT relation in Eq. (3) to CMIP data. Using the simplified FDT relation not only allows for a clean comparison with prior work, but is necessary because the full response matrices $\hat{L}$ in Eq. (1) cannot be reliably estimated from the relatively short CMIP simulations. We do not know the forcing that acts on the jet, but if we assume it aligns with the SAM in the same way across all models (as was done implicitly in e.g. Kidston and Gerber, 2010; Simpson and Polvani, 2016), we can directly relate forcing and response projections onto EOF1, $\delta\hat{u}_1 \sim \lambda_{1,1}$.

Simpson and Polvani (2016) used the jet shift instead of the EOF1 response projection $\delta\hat{u}_1$ as an additional approximation.

To test the simplified FDT relation, we plot the raw SAM timescale against the forced response in Fig. 7a. Different from Simpson and Polvani (2016), we do find a positive correlation of $r = 0.37$ ($p = 0.02$). However, this correlation is weak and dependent on a few data points: for example, removing the single data point with the largest EOF1 response reduces the correlation to $r = 0.28$ ($p = 0.09$), and indeed a Spearman's rank correlation test (less sensitive to outliers than the Pearson

correlation) gives a value of $r = 0.26$ ($p = 0.11$) when including all data points. Hence, the result in Fig. 7a shows no clear evidence of a physical relationship between SAM timescale and SAM response during austral summer.

An even smaller correlation coefficient of $r = 0.19$ ($p = 0.25$) is obtained after correcting the SAM timescale (Fig. 7b). We note however that by accounting for the stratospheric influence, we have reduced the range of SAM timescales in the models during austral summer, consistent with the findings in Fig. 5. The lack of correlation in Fig. 7b (despite using the corrected

SAM timescale) could be due to the combined effects of stratospheric influence on the response, and varying amounts of warming among the models. Ceppi and Shepherd (2019) showed that changes in mean VB date between historical and RCP8.5

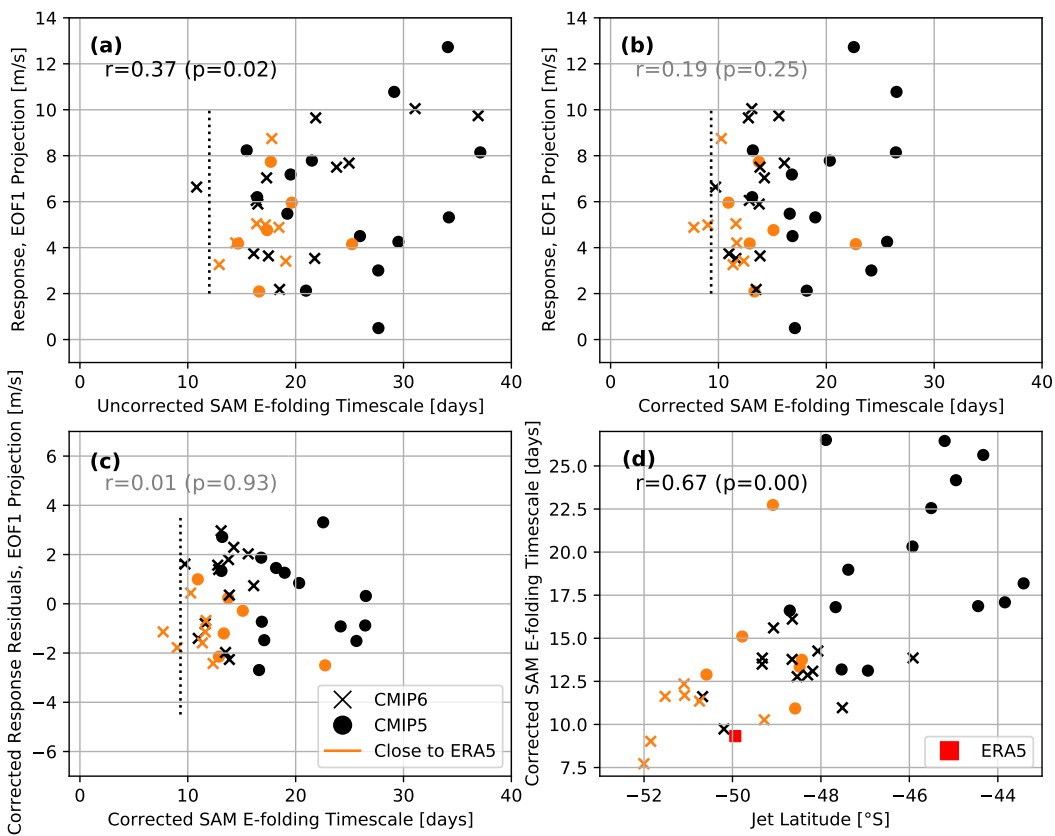

**Figure 7. (a)** CMIP5&6 850 hPa zonal wind response in DJF, projected onto EOF1, plotted against SAM timescale. **(b)** As in (a), but after the VB influence has been regressed out of the SAM timescale. **(c)** As in (b), but the impacts of the stratospheric VB response and differences in global warming have been regressed out of the EOF1 response (see text for details). **(d)** Corrected SAM timescale plotted against climatological jet position. Orange symbols denote models whose climatological jet latitude never deviates further than $5°$ from ERA5 over the year, while vertical dotted lines show the ERA5 DJF SAM timescale. Correlations with $p > 0.05$ are greyed out.

scenarios among the CMIP5 models explain a substantial fraction of the spread in DJF austral jet responses. This is because a delayed VB date leads to a delayed jet shift, meaning the jet spends on average less time in the equatorward state. This response is the result of forcing external to the troposphere (on the timescales considered here), so we do not expect it to be captured by our FDT approach based on tropospheric internal variability. Additionally, the different amounts of warming observed in the models will lead to different response strengths. We thus use multilinear regression to regress out the effects of changes in mean VB date and mean global temperature on the EOF1 response, and describe the result as the "corrected residual response". This residual response is plotted against the corrected SAM timescale in Fig. 7c.

Even after regressing out the VB influence from both SAM timescale and jet response (plus the effect of global warming from the latter), the relationship between SAM timescale and response in Fig. 7c remains non-existent. Interestingly, however, we find that the six CMIP5 and seven CMIP6 models that are defined as being close to the ERA5 climatology (i.e. whose jet position is within $5°$ of ERA5 throughout the year after applying a 20-day running window) have corrected SAM timescales very similar to one another (except for ACCESS1-3 in CMIP5) and also, while slightly too large, similar to the ERA5 corrected timescale. This supports the idea that climatological jet latitude is an important control of the SAM timescale, which is in agreement with the correlation between climatological jet position and corrected SAM timescale shown in Fig. 7d. (Note that a lower correlation of $r = 0.49$ is found if the uncorrected SAM timescale is used instead, suggesting the stratospheric influence partially obscures the relationship between jet latitude and SAM timescale). The result is also consistent with the previously mentioned findings of more equatorward jets having longer SAM timescales (Kidston and Gerber, 2010; Barnes and Hartmann, 2010). Additionally, the models closer to ERA5 also exhibit a smaller spread in forced response. We therefore find tentative evidence that these models might capture the forced response better than others, although future work should examine this possibility in a more systematic way.

There are several possible explanations for the lack of a relationship between corrected EOF1 timescale and response in Fig. 7c. One possibility is that the FDT formulation of Gritsun and Branstator (2007) used here is not applicable in this system; e.g. the assumption of Gaussianity has been shown to be problematic, as noted in Appendix A. Yet even if we assume the FDT formulation used here is adequate, there are still other possible explanations; for example, if the forcing does not project strongly onto EOF1 then the EOF1 timescale may not be important for the response prediction, even if it is the largest entry in the response matrix $\hat{L}$ (Eq. 2). Even if the forcing does project strongly onto EOF1, other terms in Eq. (2) could still be important, making the one-dimensional relation of Eq. (3) an over-simplification; a related issue has been raised by Hassan-zadeh and Kuang (2016). Note however that we did not find improved results when including the EOF2–EOF1 interaction (not shown). Yet another issue is the simplification that the forcing $\delta \boldsymbol{f}$ on the jet is similar across models: for example, changes in meridional temperature gradient, which determine the changes in midlatitude baroclinicity and thus force the jet, are highly model dependent (Ceppi et al., 2014; Harvey et al., 2014). Repeating the analysis with prescribed sea surface temperature CMIP simulations did not improve results, however (not shown). Whatever the reason, our results show that even when taking the stratospheric influence on the troposphere into account, the SAM timescale alone is insufficient to provide an emergent constraint in austral summer on CMIP jet shift projections.

## 5  Summary and Conclusion

We demonstrate that interannual stratospheric vortex breakdown (VB) variability approximately doubles the persistence timescale of the Southern Annular Mode (SAM) during austral summer in the ERA5 reanalysis as well as in CMIP5&6 models, consistent with the interpretation of Byrne et al. (2017). We use a barotropic model to mechanistically demonstrate and quantify this effect, which we mimic using a simple torque forcing. Once VB variability is accounted for, the seasonally enhanced SAM persistence in ERA5 almost completely vanishes. In further support of the barotropic and ERA5 results, CMIP models also exhibit an approximate halving of the SAM timescale when correcting for the impact of VB variability. This helps to quantify the stratospheric contribution to the summer SAM timescale increase (e.g. Simpson et al., 2011, 2013).

Barotropic model results demonstrate how the inflated SAM timescale leads to an overestimation of the predicted forced response when using fluctuation-dissipation theory (FDT). After removing the stratospheric influence, we obtain quantitatively correct FDT predictions for the barotropic model. Extending these results to ERA5, the reduced SAM timescale suggests we should not expect a significantly larger jet response in austral summer compared to other seasons. However, the SAM timescale alone cannot explain inter-model differences in the forced jet response across CMIP models, even after accounting for stratospheric influence and difference in warming. This shows that at least in austral summer no simple one-dimensional relation between SAM timescale and jet response exists. Nevertheless, correcting for the stratospheric influence does substantially reduce the model spread in both SAM timescale and forced response, especially when only considering models that closely follow the ERA5 climatology. While we ultimately did not find an emergent constraint, our analysis helps to clarify the discrepancy in seasonality of SAM timescale and forced response found by Simpson and Polvani (2016).

We speculate that better results may be obtained when using a different formulation of FDT that does not rely on the Gaussianity assumption (e.g. Cooper and Haynes (2011)). Another approach could be to include a suitable stratospheric representation into the state space, as multivariate FDT formulations can improve predictions (e.g. Fuchs et al. (2015)), or to model the influence of the stratosphere onto the troposphere using a stochastic FDT approximation (Majda et al., 2010). However, it is presently unclear whether these approaches could be used practically for identifying emergent constraints in an inter-model comparison of GCMs, given their complexity.

## Appendix A:  FDT Issues

Several studies have investigated potential issues with the application of FDT to the climate system. One is the assumption of Gaussianity of the state vector. Both Majda et al. (2010) and Cooper and Haynes (2011) showed an improved performance in simple models when using a formulation of FDT that does not rely on the assumption of Gaussianity. However, their methods are complex and therefore do not lend themselves easily to inter-model comparisons. Practical numerical issues were addressed as well, such as the choice of a reduced state space (Majda et al., 2010; Fuchs et al., 2015; Gritsun and Branstator, 2016), problems that can arise from the state space truncation (Hassanzadeh and Kuang, 2016) or the choice of a numerically advantageous basis (Cooper et al., 2013).

## Appendix B: EOF-based FDT

Here we show the derivation of the EOF-based FDT method used to make the predictions in Fig. 6. We also explain why for
our use case it is more interpretable, numerically efficient and stable to make FDT predictions in EOF space.

### B1 Derivation in EOF Basis

We start from the FDT formulation derived by Gritsun and Branstator (2007). They consider the state vector $\boldsymbol{u}$ of a system,
which is then perturbed by a forcing $\delta\boldsymbol{f}$. The response is the difference in the two state vectors $\delta\boldsymbol{u} = \mathbb{E}[\boldsymbol{u}' - \boldsymbol{u}]$. They find a
link between forcing and response via

$$\delta\boldsymbol{u} = L\delta\boldsymbol{f}. \tag{B1}$$

The matrix $L$ is given by

$$L = \int\limits_0^\infty C(\tau)C(0)^{-1}\mathrm{d}\tau, \tag{B2}$$

with the covariance matrix $C(\tau) = \mathbb{E}[\boldsymbol{u}(t+\tau)\boldsymbol{u}^T(t)]$, where Gritsun and Branstator (2007) assume $\mathbb{E}[u] = 0$ for simplicity, but
the results hold for non-zero average. Several additional assumptions were made for which we direct the reader to Gritsun and
Branstator (2007).

We now wish to transform this result into a basis given by the EOFs. To perform the transformation we insert the singular
value decomposition (SVD) of our state vector into Eq. (B2). The SVD is given by $\boldsymbol{u}(t) = U\Sigma V^T(t)$, with the EOFs as the
columns of $U$, the corresponding principal components the columns of $V$, and $\Sigma$ containing the singular values $\sigma_i$ on the
diagonal. The calculation is partly similar to what can be found in Sheshadri et al. (2018) and is left to the reader. Denoting a
quantity given in the EOF basis with $\hat{\phantom{x}}$, we find [1]

$$\delta\hat{\boldsymbol{u}} = \hat{L}\delta\hat{\boldsymbol{f}}. \tag{B3}$$

The response matrix in EOF space $\hat{L}$ is given by

$$\hat{L} = \Sigma\int\limits_0^\infty \hat{C}(\tau)\mathrm{d}\tau\,\Sigma^{-1}, \tag{B4}$$

with $\hat{C}(\tau) = \mathbb{E}[V^T(t+\tau)V(t)]$. A more interpretable way of writing Eq. (B4) is given by

$$\hat{L} = \begin{pmatrix} \lambda_{1,1} & \frac{\sigma_1}{\sigma_2}\cdot\lambda_{1,2} & \cdots & \frac{\sigma_1}{\sigma_n}\cdot\lambda_{1,n} \\ \frac{\sigma_2}{\sigma_1}\cdot\lambda_{2,1} & \lambda_{2,2} & \cdots & \frac{\sigma_2}{\sigma_n}\cdot\lambda_{2,n} \\ \vdots & \vdots & \ddots & \vdots \\ \frac{\sigma_n}{\sigma_1}\cdot\lambda_{n,1} & \frac{\sigma_n}{\sigma_2}\cdot\lambda_{n,2} & \cdots & \lambda_{n,n} \end{pmatrix}, \quad \lambda_{i,j} = \int\limits_0^\infty \mathbb{E}[V_i(t+\tau)V_j(t)]\mathrm{d}\tau, \tag{B5}$$

---

[1] This formulation does not include weighting; if one wants to use a weighting, e.g. the commonly used $\sqrt{\cos\theta}$, the relationship becomes $\widehat{N\delta u} = \widehat{L(N\boldsymbol{u})}\widehat{N\delta f}$, with $N$ a diagonal matrix containing the weights and the EOF space spanned by $U(N\boldsymbol{u})$

with $\lambda_{i,j}$ the integral timescales of the lagged correlations between the principal components $i$ and $j$.

## B2 Interpretation

We now wish to make the results in Eq. (B5) more interpretable. We note that the main diagonal of $\hat{L}$ contains the autocorrelation of the respective principal component, which is the same result as was obtained by Ring and Plumb (2008), although they used Principal Oscillation Patterns (POPs) instead of EOFs as basis functions. We also find off-diagonal terms, which are given by the cross-correlation of the PCs multiplied by the ratio of the singular values, which is the same as the ratio of the explained variance of the two modes.

Additionally, we note that the upper left entry $\lambda_{1,1}$ is the decorrelation timescale of the first EOF, equivalent to the SAM timescale. This is the basis for approximating Eq. (B4) as Eq. (3). The validity of this approximation depends on what the exact forcing is and also how strong the off-diagonal cross-correlation terms are.

An easy way to grasp the importance of different modes and their interactions in forming the response is by visualising the matrix $\hat{L}$. We show this for the barotropic experiment setups VB-FIX and VB-VAR in Figs. B1(a–b), with the statevector $\boldsymbol{u}$ the zonally averaged zonal wind. We find positive entries on the main diagonal: mathematically, this is because the entries are given by the auto-correlation timescales, which are usually positive; physically this means that forcing a mode will create a positive response in it. Nevertheless, due to negative off-diagonal entries in $\hat{L}$ negative responses can still occur; for example, Figs. B1(a–b) show a negative $\hat{L}_{2,1}$ entry, which means that forcing EOF1 will lead to a negative EOF2 response.

We can also quantify the impact of interannual VB variability on $\hat{L}$ by taking the differences between the response matrices with and without this effect (Fig. B1(c)). In line with the other findings in this paper, we see that the entry $L_{1,1}$ is especially affected, corresponding to the SAM timescale. Furthermore, other off-diagonal entries also appear inflated, particularly those in the first row, representing the interaction of higher-order EOFs with EOF1. This shows again that only considering $\hat{L}_{1,1}$ might be an over-approximation of Eq. (2).

## B3 Challenges & Advantages

We note that in the upper right part of $\hat{L}$ the singular value ratios can become very large by construction. This can pose a practical problem when working with a limited dataset. The principal component cross-correlation timescales in the upper right are usually not well resolved in a short dataset (since the integral correlations between low and high EOFs are generally weak) and can be error dominated. Multiplying those entries with large singular value ratios can therefore amplify noise.

We were able to circumvent this problem in two ways. First, by simply taking out all the most upper right entries. While effective, this method neglects the possibility of well-resolved and important correlations between high and low order EOF's. Therefore, with the second method we try to detect error dominated entries by calculating an ensemble of $\hat{L}$ from repeatedly subsampled data. Comparing the ensemble of $\hat{L}$, we take out only those entries that show particularly large variance. Using the second method, we were able to achieve the same prediction accuracy with 1/10th of the data compared to a control experiment. Both of these methods are only possible in EOF space. To make the predictions in Fig. 6, we use the first method and restrict the response matrix to elements no more than six entries off the main diagonal.

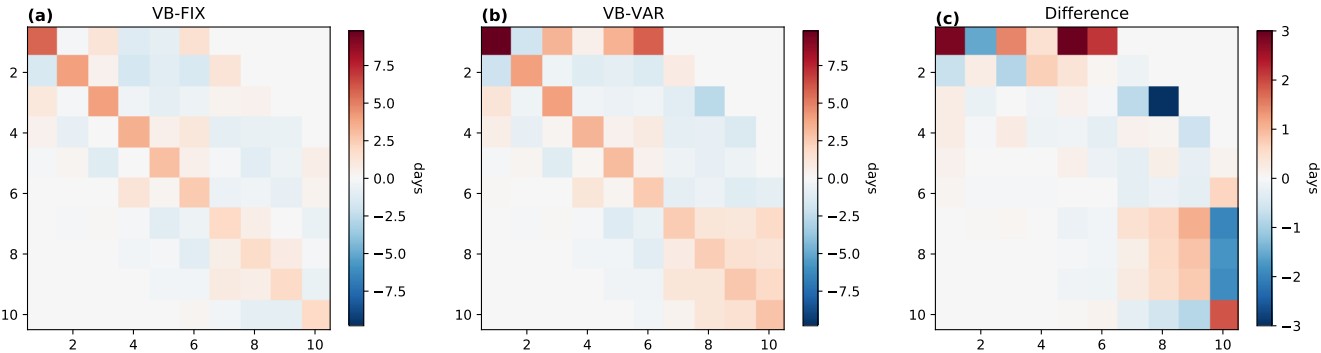

**Figure B1.** Response matrices $\hat{L}$ calculated from the barotropic setups using Eq. (B5) but for practical purposes with an upper integration limit of 40 days **(a)** VB-FIX **(b)** and VB-VAR. These are the $\hat{L}$ matrices used to make the predictions in Fig. 6, shown here in their respective EOF bases. **(c)** Difference between (a) and (b) in the EOF basis of the OFF control experiment, highlighting the impact of interannual VB variability.

A clear advantage of using the FDT method in EOF space is the reduced numerical cost. It allows for making efficient use of the dimensionality reduction, by only retaining the first $n$ EOFs in Eq. (B4), meaning we only have to calculate $n^2$ entries in $\hat{L}$.

Lastly, the method presented here might offer greater physical insight under some circumstances, compared with non-EOF based methods. In the cases where the EOFs represent a physically interpretable structure, a visualisation of $\hat{L}$, as shown in Fig. B1, will reveal for every forcing, which modes interacted how strongly to give the observed response. The latter point of course holds only if the FDT provides a satisfactory response prediction.

*Author contributions.* P.B. designed the FDT methodology, performed the model experiments and analysis, and wrote the paper. Both P.C. and T.G.S. contributed to the interpretation of the results and the writing of the paper.

*Competing interests.* We declare no competing interests.

*Acknowledgements.* We are grateful to Ed Gerber for helpful discussions. We also thank two anonymous reviewers for their constructive criticism. PB was supported by the Centre for Doctoral Training in Mathematics of Planet Earth, with funding from the UK Engineering and Physical Sciences Research Council (EPSRC) and the Department of Mathematics of the Imperial College London. PC was supported by an Imperial College Research Fellowship and NERC grant NE/T006250/1.

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
