# Peer review of "Relationship between Southern Hemispheric jet variability and forced response: the role of the stratosphere"

_Weather and Climate Dynamics, 2021_

## Author Comment (AC1)

**Final response to the reviewer comments on WCD-2021-78**

The reviewers' comments are marked in blue, while our responses are marked in black.

As a general note, we will be adding the same analysis as was done on CMIP5 models for the CMIP6 model ensemble to the paper.

**Reviewer 1**

This study helps to clarify the complex relationship among the extratropical jet climatological location, its timescale of meridional shift, and the forced responses seen in observations and models. Especially, it helps to answer (but not fully solve) two long-standing problems: 1. most GCMs overestimate the timescale of the Southern annular mode; 2. the fluctuation-dissipation theorem predicts the forced jet shift to be proportional to its timescale, but no such relationship is found among GCM simulations. This study shows that the prolonged timescale in austral summer is largely due to the variations in the breakdown date of the stratospheric polar vortex. Removing such effect leads to a closer agreement in the annual mode time scale between observations and GCMs, and a much better agreement between the forced response in a simplified barotropic model and the estimation from the fluctuation-dissipation theorem. But it does not lead to a better correlation between the timescale and the forced responses among GCMs. Overall, it is a well written manuscript with carefully designed experiments. I have a few minor comments for the authors to consider.

1. The authors very nicely showed the effect of the polar vortex breakdown date on the annual mode timescale. A perhaps slightly different way to interpret these results is that the jet is in two regime depending on whether the troposphere and the stratosphere is strongly coupled (that is, coupled in spring/autumn vs no coupling in summer/winter or torque on vs off). The fluctuation-dissipation theorem considers variations based a particular mean state, and therefore cannot count for the regime shift before and after polar vortex breakdown (VB). The authors used the regression against the VB date to count for such regime shift. It certainly works as shown in the manuscript, but I wonder if there is a cleaner way to do that. I a bit concerned that the effect of such regime shift may not be linear to the timing of the regime shift. For example, VB occurring at Nov 1 and Nov 5 may be different compared to VB occurring at Dec 1 and Dec 5. One thing that might be worth trying is to construct a conditional annual cycle: S(date of the year)=S1 if before VB, S=S2 if after VB, and calculate anomalies by removing this conditional annual cycle.

We thank the reviewer for their positive and helpful comments. We have tested the robustness of the regression approach in two ways. First by using a non-linear regression method, quadratic in this case. Second by binning the years by their VB day and calculating the SAM timescale for each bin separately (thus not assuming linearity), which we understand to be the same method as the reviewer is suggesting when using n=2. The results were not qualitatively different, neither for ERA5 nor for the CMIP models. In the plot below we show the results for ERA5. All methods show very good

agreement of the corrected SAM timescale, suggesting that non-linear effects are minor. We will make a mention of this in the revised paper.

[Figure]

2. CMIP5 models in general have a too-late VB breakdown than observations (e.g., Wilcox and Charlton-Perez 2013 JGR). I therefore expect that the VB's effect peaks at the later time of the year in CMIP5 models than in ERA5. But in Fig. 5, the lowest corrected SAM fraction in models is found at a slightly earlier date than in ERA5. This might be beyond the scope of this manuscript, but I would love to see some discussions regard this. One possibility is that the regression does not isolate the VB effect clean enough as mentioned in my last point.

We agree that a later mean VB should result in a later peak in SAM timescale reduction, according to the arguments presented in this paper. Wilcox and Charlton-Perez 2013 (WCP13) found a delay of 12.5 days between the ERA-Interim reanalysis product and a subset of 14 CMIP5 models, using the historical period of 1979-2005.

In our analysis the delay is only 1.5 days, but note that there are substantial differences in the analysis. First, we are using a wind-based method to determine the vortex breakdown, compared to the temperature-based method in WCP13; our reanalysis product is ERA5 compared to ERA-Interim; we use a larger ensemble of CMIP5 models; and lastly, we use a longer historical time period (1950-2005 for the CMIP5 models and 1950-2020 for ERA5). When using the time period 1979-2005, the delay grows to 3.3 days, though this is still smaller than what is found in WCP13.

While it might be of interest to investigate the differences in the analysis, we believe that the results presented in this paper are self-consistent.

Lastly, the figure below shows the same analysis for a set of 39 CMIP6 models (which have smaller biases than the CMIP5 models), which shows that the delay in peak SAM timescale reduction compared to ERA5 vanishes. This new result will be included in the revised paper.

[Figure]

Editorial comments:

Line 178: "dotted" -> thin

Will be changed.

Line 197-198: delete "(we only show… identical)" since both VB-VAR and VB-FIX are shown in Fig. 6.

But not the numerical response, therefore we think this should stay as is.

Fig 5a legend: it may be more meaningful to move the "CMIP5 average" to the two solid lines, rather than the title of all four lines.

We agree and will make this change.

Fig. 6: The two legend box of the same figure panel is confusing. It may be better to combine them. Also in Fig. 6a, it may be better to plotting the orange dashed line on top of the black line.

We agree with the second point and will make the change. On the first point, however, we personally find that combining the legends reduces clarity, so we suggest keeping them the way they are.

**Reviewer 2**

The authors revisit the connection between southern annular mode timescales and the response of the southern hemisphere jet to anthropogenic climate forcings. Simple arguments from the fluctuation dissipation theorem suggest that the two should be to some extent correlated in global models such that those models with a longer timescale should have a larger response. Such a relationship could be valuable as an emergent constraint on projected changes in the circulation; this is thus an important topic that should be of broad interest to the readers of WCD.

This connection has been questioned by previous studies. The present study considers whether the effects of variability in the austral stratospheric polar vortex on SAM timescales could be responsible for the apparent lack of correlation identified in CMIP5 models by Simpson and Polvani (2016). They develop a regression methodology for removing the stratospheric contribution to the timescales based on some idealized integrations of a barotropic model, and apply this to the CMIP5 archive. In the end they find that this does not improve the correlations. Hence this is essentially a negative result.

Given the potential importance of identifying such an emergent constraint, this kind of negative result may still be valuable to report. However, the authors have not done a very good job of highlighting what insights this paper adds to the literature; in fact very little previous work has been cited at all. In particular, many other studies have come to similar conclusions that applying the FDT naively in this way is often quantitatively incorrect; see, e.g.

https://doi.org/10.1016/j.physd.2016.05.010 https://doi.org/10.1175/2010JAS3633.1 https://doi.org/10.1175/JAS-D-16-0099.1

though this is by no means exhaustive.

In order to be worthy of publication, the present study should be clearly placed in the context of the existing literature, and the authors need to make a compelling case for why this negative result adds something of value to the literature. This may mostly an issue of rewriting, but I have nonetheless recommended major revisions on this basis.

We thank the reviewer for their constructive feedback. We agree that our results should be better placed in the context of existing literature, and we will revise the manuscript accordingly. We note, however, that the literature listed above focuses on general shortcomings of FDT when applied to the climate system, often studying very idealised models. We focused specifically on the SH midlatitude jet stream and the open question

of the relationship between SAM timescale and forced response. The analysis presented therefore has a different focus than the above papers.

Some further general and specific comments are given below.

General comments

Assumption that the stratosphere is exogenous

The authors have assumed without much comment that the contribution of the stratosphere to the southern annular mode timescales should be 'corrected for', in that it should not be seen as implying a stronger response to an imposed force. This is clearly the case in the barotropic model where the effects of the 'stratosphere' is explicitly an exogenous force. But it's not at all obvious that this should be the case in the real atmosphere (or in climate models). The framing (adopted from Byrne et al. 2017) is that the stratospheric influence renders the temporal statistics of the tropospheric jet 'non-stationary.' But how does this really connect to the FDT? Consider, for instance, a climate model with a well-resolved stratosphere with explicitly fixed or periodic boundary conditions. Such a dynamical system can easily include seasonal variability in the stratosphere and troposphere, including variability in the date of the vortex breakdown that leads to the DJF peak in SAM timescales. Assuming that this dynamical system has some kind of well-defined attractor, the statistics are certainly stationary after some initial transient evolution to the attractor. If we apply the FDT to this dynamical system, on what basis should we consider some parts of the system 'exogenous'?

There is some dynamical intuition that this should be the case, but it's not obvious to me how to justify this intuition in the context of the FDT, and certainly the 'non-stationary' argument can't be right. Why shouldn't the full FDT apply?

I don't expect a full answer to this question, but the authors have just tacitly assumed this point which is not at all obvious. More generally, how do we know that the stratosphere is the only 'exogenous' impact on the SAM timescale? Also, without a clear theoretical basis for this approach, how do we know this 'correction' is even the right thing to do? The barotropic model is hardly a justification.

We agree with the reviewer that the assumption of an exogenous stratospheric influence has to be discussed more clearly in the paper.

We think that by using only one zonally averaged tropospheric variable (usually zonal wind or geopotential height at one pressure level) in the state space when making FDT arguments about the jet stream, as is common in the literature (e.g. Gerber et al. 2008; Kidston and Gerber 2010; Simpson and Polvani 2016), one implicitly assumes the stratosphere to be an exogenous influence. For example, one could never hope to capture the tropospheric response to a stratospheric perturbation using a troposphere only FDT. Since much of the stratospheric variability may ultimately be of tropospheric origin, it might be possible to capture the influence of the stratospheric pathway without resolving the stratosphere explicitly. But these processes happen on longer timescales

than the tropospheric variability and require more variables to model than just zonal wind (e.g. Gupta et al. 2021), which would seem to make the hope for a simple 1D emergent constraint futile.

As an aside we note that this is not too dissimilar to the mutual influence of troposphere and sea surface temperature (SST). While much of the SST variation comes from the troposphere, the ocean has a significantly longer memory than the atmosphere does. Therefore it is common to treat SSTs as an exogenous forcing to the troposphere.

So therefore we agree that when considering a larger state space that somehow represents the stratosphere and/or tropospheric drivers of the stratosphere and is able to practically deal with multiple timescales, there might not be the need to "correct" for a stratospheric influence on the troposphere. Indeed this could be an interesting line of future research. But when considering the mid-troposphere zonally averaged zonal wind as a separate system by itself, as is often done, we believe a correction is necessary if one wants to apply FDT arguments. Ultimately we present a parsimonious model with the working assumption that the stratospheric influence is exogenous to the troposphere, but we agree that this assumption has to be clearly stated.

We agree with the reviewer that we cannot be certain that the stratosphere is the only 'exogenous' influence on annular mode timescales – indeed it seems likely that other processes could have an impact. However, given that the DJF peak in PC1 timescale is the only remarkable feature in the annual cycle (there is no peak in PC2 timescale for example), we do not see obvious signs of other strong influences.

**Barotropic model**

The authors have modelled the effects of the stratosphere in the barotropic model as a time-dependent imposed force. This will have the desired effect of modifying the jet, but the nature or origin of such a force in the real system remains a topic of some debate. Does the spatial structure of this imposed force have any effect on the ability of the regression model to correct for the timescale?

We agree that the imposed torque forcing is not an exact model of reality; we are simply using an arbitrary poleward shifting torque to demonstrate how interannual vortex variations can increase persistence timescales. We note that our proposed mechanism does not depend on a particular forcing structure; the only important aspect is that the forcing produces a poleward jet shift. This will be noted in the revised manuscript.

Since we are regressing out the VB influence from the zonally-averaged zonal wind anomalies before we calculate persistence timescales, we do not expect the exact shape of the imposed torque to have an influence on the results. Furthermore, we have verified that other torque structures, including monopoles, can also produce a poleward jet shift in the barotropic model. This is consistent with other idealised modelling evidence (e.g. Lorenz 2014).

**Use of breakup date**

While the timing of the vortex breakup is probably the single biggest source of year-to-year variability in the austral vortex, it doesn't capture everything. The regression methodology could just as easily be applied to, say, the anomalous zonal mean zonal wind at 50 hPa for instance; this would include more stratospheric variability.

We agree with the above comments but want to point out that when regressing against anomalous u50, one has to choose a somewhat arbitrary lag time, since the influence from the stratosphere to the troposphere needs time to propagate. The below plot is the same as fig. 5, but using anomalous u50 with a lag time of 7 days for the regression methodology (qualitatively similar plots are obtained for lag days 5 or 10).

Interestingly, the residual CMIP5 SAM vortex peak has vanished and we see a reduction in SAM timescale throughout the year. This suggests that the residual peak is of stratospheric origin and that the stratospheric influence is not restricted to the vortex breakdown period only, although it is strongest during that time.

Nevertheless using this method does not make any qualitative difference to the results in Fig. 6. For the sake of simplicity we therefore prefer sticking to the VB methodology, while noting in the revised manuscript that alternative metrics of vortex variability may also be used.

[Figure]

Use of EOF1 only for GCMs

At various points the claim is made that applying the 'full' FDT calculation in EOF state space would be prohibitive for the GCMs. I don't see any reason this is the case if you are already limiting the state space to the tropospheric 'barotropic' zonal mean zonal wind. The same approach to choosing a subset of EOFs could easily be applied, and some discussion about the validity of restricting the FDT to the leading mode of variability could usefully be made. This I think would be somewhat novel.

Even after limiting the state space to the zonal-mean zonal wind at 850 hPa, the data length is not sufficient to resolve the correlation structure properly. As a comparison, the barotropic model uses 2000 'years' of data, while we only have 54 years of data in the historical model runs in CMIP5. We will make a mention of this in the revised paper.

When only considering the response projection onto EOF1, the FDT prediction of equation 4 is exact (to the extent that the chosen FDT method is correct of course), equation 5 then approximates it by dropping all terms except for the first which represents the autocorrelation of EOF1. We did try to include the second term of equation 4, $\widehat{L_{1,2}}\ \widehat{\delta f_2}$ , which relates to the cross-correlation of EOF1 with EOF2, but this did not lead to better results.

Two technical questions that should be clarified in the text (not just in the response):

1) How is are the autocorrelation `timescales' calculated? The integral expressions in the appendix are more general, but only reduce to the decorrelation timescale for specific processes. Yet it is common to assume exponential decay, at least over some fixed set of lags. What was done here, and how is the choice justified?

For the barotropic experiments, the response matrices L, shown in Figure A1, are calculated using equation A4, but for practical purposes the upper limit of the integration was set to 40 days. Those response matrices have been used to make the predictions shown in Figure 6. This will be clarified in the revised text.

As outlined in l102-105, the SAM/EOF1 timescales were calculated as the e-folding timescales of the autocorrelation of PC1 of u850 anomalies, i.e. the time when the autocorrelation drops below 1/e. In the case of truly exponential decay, the e-folding timescale and integral correlation timescale are the same, as mentioned in L117-118. We decided to use the e-folding timescale since it is a very common choice in the literature and it does not require the arbitrary choice of a practical upper integration limit.

The plot below shows the same results as in Figure 5, but using the integrated correlation timescale with an upper integration limit of 80 days. While the result is noisier, the qualitative result is the same, suggesting that the choice of timescale measure does not influence the overall result too much. Interestingly, the very long timescale peaks shown by some models vanish (and do not appear for larger upper integration limits), which suggests that the decorrelation function has some deviation from an exponential form.

[Figure]

2) What is the state space used in the appendix? The full dynamical state space of the barotropic model, or just the zonal mean zonal wind?

The latter – this will be clarified.

l152: 'position decreases' -> 'latitude moves equatorward'

Changed

Fig. 6: Are the dot-dashed lines in Fig. 6c and d the same as the full lines in Fig 6a and b? The figure caption suggests so, but the label suggests otherwise, and they don't look the same for the Gaussian case. It would be particularly helpful to clearly evaluate the importance of the EOF1 assumption.

The lines are not the same. The full lines in Fig. 6a and b show the total numerical response to the EOF1 type and Gaussian Type forcing respectively. The dot-dashed lines show the EOF1 projection of the numerical response, as stated in the caption.

In fact this plot was intended to help clarify the EOF1 assumption (as mentioned in L210-215), as it demonstrates the impact of using the 'full' FDT method to predict the full response to a forcing, as was done in 6a and b, versus using only the SAM timescale to predict the EOF1 component of the response, as was done in 6c and d.

l232: I'm not sure that constrained is the right word here - there's no validation of the current result. The range of timescales has been reduced however. A similar comment applies to l280.

We will change it to 'reduced'.

Fig. 7: Some comments are made highlighting the fact that models with a climatological jet position that is close to ERA5 have timescales that are generally close to the reanalysis as well. Is this agreement better for 'corrected' timescales than it is for the uncorrected timescales? This is not shown.

One can see the difference by comparing the orange triangles relative to the dotted line in Figure 7a and b for uncorrected and corrected timescales respectively. Qualitatively, we find that the difference in timescale of these models with ERA5 is 6.5 days or 53% for the uncorrected case, and 5.5 days or 58% in the corrected case. So while the absolute difference to the ERA5 timescale decreased after the correction, the relative difference was slightly increased. However, our point was that these models are closer to ERA5 in both climatology and timescale relative to the whole ensemble, and that their spread in corrected response is also smaller than that of the whole ensemble. This suggests that they possibly capture the dynamics better.

l286-287: This has already been done by a number of studies (see those references cited above, for instance).

This part will be placed better into existing literature as mentioned above.